## [Transparent Peer Review file · Nature Communications]

The genetic driver of Acute Necrotizing Encephalopathy, RANBP2, regulates the inflammatory response to Influenza A virus infection

Corresponding Author: Dr Nathalie Arhel

Version 0:

Reviewer comments:

Reviewer #1

(Remarks to the Author)

Desgraupes et al. addressed the molecular basis of how a clinically relevant mutation in RANBP2 can promote the induction of acute necrotizing 1 encephalopathy. Although RANBP2 knockdown in A549 cells had no effect on overall IAV propagation, it resulted in a significant increase of viral negative-sense genomic vRNA and positive-sense cRNA sequences. An increase in cytoplasmic NP and the co-localisation of cytoplasmic RANBP2 with cytoplasmic NP with PB1 further indicated that RANBP2 knockdown induces an increase in the export of vRNPs to the cytoplasm. Interestingly, RANBP2 knockdown neither affected internalization nor nuclear import of IAV, but it facilitated the import of newly synthesised polymerase. Analysis of the stoichiometry of viral segments indicated that depletion of RANBP2 increases the replication of all vRNA segments, whereas it disproportionately affects their export back to the cytoplasm. Knockdown of RANBP2 in monocyte-derived macrophages resulted in enhanced proinflammatory cytokine and chemokine responses. U2OS cells carrying the RANBP2-T585M mutation homozygously showed enhanced CXCL10 expression following IAV infection. A similar effect was detected upon transfection of a constitutively active RIG-I variant. In WT clones, RANBP2 localised to the nuclear envelope, whereas clones homozygous for RANBP2-T585M lost the characteristic nuclear rim labelling indicating that the RANBP2 mutation disrupts RANBP2 retention at the nuclear envelope, whereas the integrity of nuclear pores seemed not be affected. In conclusion, the clinically relevant RANBP2 mutation facilitates the re-import of polymerase into the nucleus, which increases viral cRNA and vRNA, and selectively exports vRNA into the cytoplasm. The dysregulation of both processes results in an unusual accumulation of vRNA segments in the cytoplasm, that serve like pathogen-associated molecular patterns that trigger enhanced cytokine responses.

The authors present a precisely planned study. The manuscript guides the reader through a series of logically designed experiments, of which each one reveals interesting results. In the end, a rather anti-intuitive mechanism is concluded how a mutation in RANBP2 can cause detrimentally enhanced cytokine responses. The paper is a logical exercise at its best. The manuscript is very carefully written by using precise and easily understandable language.

Unfortunately, the figures and figure captions are not as carefully prepared and need some improvement.

Minor points:

When the predominant ANE1-associated mutation c.C1754T: p.T585M is mentioned first, the nomenclature should be briefly introduced. Amongst the broad readership of the journal, not everybody is familiar how genomic alterations are identified. In Fig. 4C details are nicely illustrated, but insufficient explanations are given. This should be done ideally in the figure legend or in the main text.

Within Figure 1, many different font sizes were used for labelings. This is impairing the readability of the figure.

Fig. 1A-C: What do single dots mean? Do they represent single values or single experiments? How frequently the experiments were repeated?

Fig. 1C: It is difficult to see where the notion comes from that in THP-1 cells an up to 40-fold increase of M1 RNA was

detected.

Fig. 1G: NTD presumably stands for N terminal domain. This abbreviation was introduced in the introduction, but not in the figure legend. This should be corrected.

Fig. 1I: Colocalization in the shRNA Control and RanBP2 groups look very similar. Some other way of visualization should be provided.

Fig. 1J: It is not detailed neither in the results section nor in the figure caption what is shown in the upper panel.

In the caption of Figure 1, a typo should be correct (J).

In Fig. 3A it looks as if the analysis of CXCL10 transcripts in monocyte-derived macrophages has been performed only twice. How can statistics be performed with 2 values? This needs re-evaluation. The same is true for the analysis of TNFalpha in monocyte-derived macrophages. How exactly the experiment was carried out? When for the IL-1beta analysis in monocyte-derived macrophages three values were analyzed, then the samples should be basically available for the analysis of the other cytokine RNAs. In the experiment with THP-1 cells, treatment with control shRNA was also carried out only twice, whereas the experiment with RanPB2 was performed three times. This is no basis to perform statistics. On a first glance, only for the studies with A549 cells enough data points were retrieved to perform reasonable statistics. It is recommended that a professional statistician cross-checks the calculations.

For Fig. 3D more details should be provided. Was the experiment carried out only once?

In Fig. 4A the caption should say: Schematic depiction of ... and in the figure this text should be deleted. Within this figure part, only one font size should be used. The labelling within the box is so small that is it hardly readable.

Fig. 4D: Sanger sequencing goes back to Fred Sanger, therefore it should be written with a capital first letter.

In the legend of Fig. 4 reference is given to (I), but (I) is not shown the figure. This should be corrected.

In Fig. 4 too many different font sizes are used. In the end, every label is important, otherwise it would not have been shown. Therefore, the authors should consider using only one font size.

Reviewer #2

(Remarks to the Author)

Summary:

In this work, entitled 'The genetic driver of Acute Necrotizing Encephalopathy, RANBP2, regulates the inflammatory response to Influenza A virus infection', Desgraupes et al. describe that loss of RANBP2 leads to increased IAV vRNA replication and increased localization of vRNPs to the cytoplasm. This is accompanied with an increased pro-inflammatory cytokine response. CRISPR-edited cells expressing RANBP2 T585M, the SNP that is observed in ANE patients, also show increased CXCL10 expression in response to IAV infection or after stimulation with RIG-I 2CARD. Additionally, the T585M mutation disrupts the localization of RANBP2 at the nuclear pore. This study shows interesting insights into the function of RANBP2 in influenza virus infection, and how the RANBP2-T585M mutation could contribute to ANE. However, it is not clear if the observed increased inflammatory response in the absence of functional RANBP2 is caused by increased viral RNA replication or by exacerbated signaling downstream of RIG-I.

Major comments:

1. All experiments shown in Figures 1-3 were done using only one shRNA. To exclude potential off-target effects of the used shRNA, key experiments need to be repeated with additional shRNAs and/or including rescue experiments (expression of shRNA resistant RANBP2).
2. Figure S1: The western blot shows the knockdown at protein level in A549 cells at 72h post lentiviral transduction. Is this the time point that was used for infection with IAV? I could not find this information in the method section. The authors should add this information and potentially provide a western blot for the time point of infection. The western blot showing RANBP2 reduction at protein level should also be added for MDM cells (where currently only a qPCR at 24h is shown).
3. Fig 1B and Fig 3A: The data showing the knockdown efficiency of shRNA-transduced THP1 cells is missing. The authors should provide a western blot showing RANBP2 levels like for the A549 cells in Figure S1B.
4. The authors state that nuclear export of segment 1, 2, 3, 7 and 8 is increased by RANBP2 KD. I find it difficult to see this in the data shown in Figure 2F. For all segments, the cytoplasmic levels seem to be increased to at least some extend, and the nuclear levels are either equal or slightly increased compared to the control. The increase of RNA levels in the cytoplasm in RANBP2 KD compared to control is between 1.5-3-fold for all segments, which is a difference of 2-fold in-between the different segments. In a qPCR assay, this change corresponds to only 1 Cycle difference. I think that from this data it is difficult to conclude that the different segments are actually affected differently by RANBP2 KD. Additionally, I wonder why there is no decrease in the nuclear RNA levels observed if nuclear export is increased.
5. While all virological assays were done in A549 cells, the experiments investigating the effect of RANBP2-T585M (Figure 4), were done in U2OS cells. It is not clear why the authors switch to a different cell line. I assume this is for technical reasons, as it might be difficult to do CRISPR editing in A549 cells. I think the authors should reproduce some of the shRNA KD data (increased cytoplasmic export of vRNPs and increased cytokine production in RANBP KD conditions) in U2OS cells to show if this cell model shows the same phenotype as A549 cells. Additionally, the increase in cytokine induction

appears much stronger in U2OS RANBP2-T585M cells (Figure 4E) compared to A549 RANBP2 KD cells (Figure 3A). Is this due to differences in the cell lines, or does the T585M mutation increase cytokine production compared to RANBP2 KD?

6. In Figure 4, the authors show an effect on CXCL10 with RIG-I 2CARD expression in RANBP2-T585M cells. Does this mean that the phenotype observed in cells expressing RANBP2-T585M is a different one than observed in the KD cells, caused downstream of RIG-I sensing and not by increased viral RNA levels or changed vRNA localization? To clarify this, I would suggest to test if import of viral polymerase and vRNP export are affected in U2OS RANBP2-T585M cells. And vice versa, could the authors repeat the experiment shown in Figure 3A with RIG-I 2CARD transfection instead of IAV infection?

7. Are the protein levels of RANBP2-T585M the same as WT RANBP2 in U2OS cells? It would be important to show this, to exclude that the observed phenotype of the T585M mutation is not caused by reduced protein levels.

Minor comments

- What is the unit of the MOIs stated in the manuscript? PFU/cell or TCID50/cell?
- I think the qPCR primers used for the validation of strand-specific RT shown in Fig S2 are not listed in the supplement. If I understood correctly, these primers are different from the qPCR primers used in the other assays, since they are specific for the tags added during the RT?
- Figure 1J: From which condition is the magnified example image and the analysis? Could the authors provide this analysis showing the signal distribution for all conditions? Could the authors additionally provide a quantification of the colocalization (e.g. pearson coefficient) for all conditions?
- Number of biological replicates is not stated for the experiments. Especially for the western blots shown in Fig 2A and 2B, how often was this experiment done?
- Figure 2E: It is not clear to me why the authors analyze nuclear export of cRNAs. As far as I know, cRNAs would not need to be exported to the cytoplasm during the IAV life cycle, since they serve as template for vRNA synthesis that occurs in the nucleus.
- Figure 2E and F: How many replicates were done of this experiment? Does each dot represent a technical replicate from the same experiment or a biological replicate from a different experiment? Please add this information to the figure legend. Additionally, the statistical analysis for this experiment should be added.
- In the qPCRs in Figure 2E and F, the data was normalized in different ways. In 2E (cRNA), normalization is relative to the nuclear fraction of the control, which means that nuclear and cytoplasmic fractions can be compared to each other, and it is apparent that most of the cRNA is located to the nucleus. In Figure 2F, the same experiment is shown for vRNA, but here the fractions are normalized to the control condition within each fraction. Thus, it is not apparent how much RNA is exported to the nucleus at the observed time point. It would be better to do the normalization in the same way as for Figure 2E.
- Line 253: 'In particular, RANBP2 was not involved in the initial export of viral transcripts, which involves the general mRNA export pathway via NXF1:NXT' – Did the authors test this? Please show data on viral mRNAs, or remove this statement from the discussion.

Comments on Methods part.

The following information missing in the methods part:

- The amounts of plasmids transfected to produce lentiviral particles.
- Detailed description of how A549 cells were transduced with the lentiviral particles. It is not clear from the manuscript if there was a selection step and if stable cell lines were produced, or if the A549 cells were freshly transduced and immediately infected with IAV in each experiment.
- Information on IAV-WSN stock production and stock titration
- Line 315: 'For experiments with NF- κ B canonical pathway inhibition, cells were pre-treated with 2% FBS culture medium containing 20 μ M of PS1145 (MedChemExpress) for 1 h at 37°C'. It is not clear which experiments were performed with inhibition of NF- κ B.

Reviewer #3

(Remarks to the Author)

The submitted study by Desgraupes et al. aims to understand the role of RANBP2 during influenza A virus infection. Through their interesting work, these authors were able to show that RANBP2 is able to directly influence the inflammatory response of epithelial and immune cells during infection. The manuscript is extremely well-written and their experimental work is very close-knit. This study provides significant groundwork towards the understanding of the exact mechanisms of RANBP2, especially in the context of patients with the genetic predisposition to developing ANE.

Major comments:

- Lines 183-185, there is an inherently large jump here from the previous section. Some improvement for the motivation here to align the two sections is missing. Moreover, it was not established if the difference in vRNAs and vRNPs dynamics were also observed in these MDMs, before looking into the expression of proinflammatory genes.
- The selection of the respiratory epithelial cell line A549 cells (adenocarcinoma) for the initial investigation makes sense. However, this reviewer would like to see the dynamics of proinflammatory gene expression after infection in upper respiratory cells to explicitly show the effect of the RANBP2 in IAV appropriate target cells. This would make an interesting comparison with the effect of infection of MDMs in this study (epithelial vs. immune proinflammatory responses).
- Figure 3, the authors show that there are differences in the gene expression pattern of these proinflammatory markers after IAV infection after an overnight infection strategy. This reviewer thinks that a variable timelapse analysis may be necessary to clearly say that the differences in gene expression are a result of the cell types. It could be possible that some gene

expression changes had happened a lot earlier in some cells and later in others. A time-resolved infection strategy with a qRT-PCR readout would help here.

- Have the authors looked into interferon stimulated genes (ISGs) and the influence of RANBP2 here?

Minor comments:

- In line 105, the figures do not match the cell lines in the text.
- Lines 178-180, well-written but must be revised to be more explicit that this is still heavily speculative at this point.
- Figure 4D, please add the clone number on the traces to make it easier to read.
- Line 765, please use the proper β symbol for IL-1 β

Version 2:

Reviewer comments:

Reviewer #1

(Remarks to the Author)

Desgraupes et al. followed many of the reviewers' recommendations and implemented several changes in the revised version of their manuscript. New controls were included, and experiments were repeated where needed in order to be able to provide statistics. Thanks to the extended discussion, more context is provided. However, in all figures still different font sizes are used which makes it difficult to read the figures. Since many readers certainly will want to verify some of the conclusions by inspection of the actual data, the authors should try to present the data in an easily readable manner.

Minor points:

Abbreviations in context of RNA are inconsistently used. In line 47 of the revised manuscript, it is defined that vRNA stands for "negative-sense single-stranded viral RNA". However, many times later in the manuscript the authors again refer to „viral RNA“. Do they use "vRNA" and "viral RNA" in a synonymously manner? If this is the case, the authors should better stay with the abbreviation vRNA throughout the manuscript after introduction the abbreviation.

shRNA is not defined in the manuscript. Considering that it is mentioned more than 100 times in the text, the authors should introduce the abbreviation, which presumably stands for "short" or "small hairpin RNA".

The expression „pro-inflammatory M1-like macrophages“ can be misleading. It is recommended that the authors better refer to "GM-CSF macrophages".

Reviewer #2

(Remarks to the Author)

In the revised version of the manuscript 'The genetic driver of Acute Necrotizing Encephalopathy, RANBP2, regulates the inflammatory response to Influenza A virus infection' the authors improved the initial manuscript. Importantly, they added RANBP2 knockdown data in U2OS cells, the cell line which is also used to characterize the ANE-associated T585M mutation, and confirmed the observed phenotype using a second shRNA for RANBP2 knockdown.

The authors removed the RIG-I data, which was rather weak in the previous version. In the revised manuscript, the authors do not show a link between increased cytoplasmic vRNA and RIG-I sensing, but I feel it is justified to make this connection from the literature.

In the revised version of the manuscript, the authors addressed all of my concerns appropriately. I have two comments regarding the presentation of the data:

- It is difficult to compare the data in Fig 3A and Fig S11, which show cytokine expression for RANBP2 knockdown with two different shRNAs in A549 cells, since the normalization of the data was performed differently. Especially for CXCL10, it looks like with shRNA#1 (Fig 3A), the induction is ~2 fold (IAV infected, control vs. RANBP2 shRNA), while with shRNA#2 (Fig S11), it is more than 50-fold. Is there a way to improve the visualization to make it easier to compare the two different shRNAs?

- Fig 2D: Figure legend states n = 5, but for PA only 4 data points are shown. Can the 5th experiment also be included?

Reviewer #3

(Remarks to the Author)

The revised study by Desgraupes et al. is a significant improvement from the prior submission. This reviewer thanks the authors for the additional data they have provided (in the manuscript and those in the rebuttal only).

Major comment:

There is only one point that this reviewer would like to raise which is the confidence on the inflammatory effect of RANBP2. The hint of its presence is there but the wide spread of measurements in Fig3A and Fig4G can raise some concerns.

Fig 3A qPCRs - measurements for control and RANBP2 have a high spread and most significant changes are being driven by 1 measurement point. Have these been tested for outliers or maybe the experiment needs to be repeated?

Fig4G- needs the dots on the box whisker plots. The spread of these measurements are extremely high. The hint of the effect is there but additional confirmation of this would be nice. Similar to Fig3A, test for outliers and a potential repeat of the experiment might be necessary.

Fig3C - LUMINEX assays do not show TNFa, IL6 and IL1b increase and only for CXCL10 which is different from what was found in A549 cells and U2OS. Any point of discussion here?

Version 3:

Reviewer comments:

Reviewer #3

(Remarks to the Author)

This reviewer would like to thank the authors for their detailed responses regarding the concerns of all reviewers. No further comments or edits the careful phrasing of line 289-293 is sufficient.

Answers to the reviewers' comments

We thank all reviewers for their thorough evaluation of our work and constructive comments.

To address the reviewers' comments, we replicated the key experiments using alternative assays or shRNA, and we performed new experiments. In particular, following the reviewer 2's suggestion to better characterize the RANBP2-T585M CRISPR clones, Figure 4 now includes evidence that the RANBP2-T585M point mutant increases IAV vRNA and innate immune signaling similarly to RANBP2 knockdown.

Overall, the revised manuscript includes **41 new or substantially revised panels** (Figs. 1B–C, 1I–K, 2A–E, 2H–J, 3A–B, 4E–H, 4J, S1A–C, S4A–D, S5A–D, S9, S10A–C, S11, S14A–B, S16, S17A–B). The author order has been updated to reflect the contributions made during the revision period with A. Decorsière moved to second author position.

Reviewer #1 (Remarks to the Author):

Desgraupes et al. addressed the molecular basis of how a clinically relevant mutation in RANBP2 can promote the induction of acute necrotizing 1 encephalopathy. Although RANBP2 knockdown in A549 cells had no effect on overall IAV propagation, it resulted in a significant increase of viral negative-sense genomic vRNA and positive-sense cRNA sequences. An increase in cytoplasmic NP and the co-localisation of cytoplasmic RANBP2 with cytoplasmic NP with PB1 further indicated that RANBP2 knockdown induces an increase in the export of vRNPs to the cytoplasm. Interestingly, RANBP2 knockdown neither affected internalization nor nuclear import of IAV, but it facilitated the import of newly synthesised polymerase. Analysis of the stoichiometry of viral segments indicated that depletion of RANBP2 increases the replication of all vRNA segments, whereas it disproportionately affects their export back to the cytoplasm. Knockdown of RANBP2 in monocyte-derived macrophages resulted in enhanced proinflammatory cytokine and chemokine responses. U2OS cells carrying the RANBP2-T585M mutation homozygously showed enhanced CXCL10 expression following IAV infection. A similar effect was detected upon transfection of a constitutively active RIG-I variant. In WT clones, RANBP2 localised to the nuclear envelope, whereas clones homozygous for RANBP2-T585M lost the characteristic nuclear rim labelling indicating that the RANBP2 mutation disrupts RANBP2 retention at the nuclear envelope, whereas the integrity of nuclear pores seemed not be affected. In conclusion, the clinically relevant RANBP2 mutation facilitates the re-import of polymerase into the nucleus, which increases viral cRNA and vRNA, and selectively exports vRNA into the cytoplasm. The dysregulation of both processes results in an unusual accumulation of vRNA segments in the cytoplasm, that serve like pathogen-associated molecular patterns that trigger enhanced cytokine responses.

The authors present a precisely planned study. The manuscript guides the reader through a series of logically designed experiments, of which each one reveals interesting results. In the end, a rather anti-intuitive mechanism is concluded how a mutation in RANBP2 can cause detrimentally enhanced cytokine responses. The paper is a logical exercise at its best. The manuscript is very carefully written by using precise and easily understandable language.

Unfortunately, the figures and figure captions are not as carefully prepared and need some improvement.

We have reviewed all the figure captions and harmonized the fonts and sizes, particularly regarding the graph axes.

Minor points:

When the predominant ANE1-associated mutation c.C1754T: p.T585M is mentioned first, the nomenclature should be briefly introduced. Amongst the broad readership of the journal, not everybody is familiar how genomic alterations are identified.

The mutation nomenclature has been introduced as follows:

(lines 75-76): “c.C1754T on the coding DNA reference sequence, leading to p.T585M at the protein level”

In Fig. 4C details are nicely illustrated, but insufficient explanations are given. This should be done ideally in the figure legend or in the main text.

Explanation has been added in the figure legend.

(lines 765-766): “This mutation on DNA (c.C1754T) leads to the T585M mutation on the RANBP2 protein (p.T585M)”

Within Figure 1, many different font sizes were used for labelings. This is impairing the readability of the figure.

This has been corrected.

Fig. 1A-C: What do single dots mean? Do they represent single values or single experiments? How frequently the experiments were repeated?

All experiments were performed 3 times. In all graphs, dots represent technical replicates from all independent experiments. This was added to the figure legend as follows:

(line 699-700): “For panels A-F, dots represent technical replicates from all independent experiments.”

Fig. 1C: It is difficult so see where the notion comes from that in THP-1 cells an up to 40-fold increase of M1 RNA was detected.

The original THP-1 panel was a mean of 2 experiments, therefore we replicated these experiments. Results in Figure 1C, which are the mean of 3 independent experiments, no longer show a 40-fold increase but rather an average of 10-fold. Therefore, this sentence was removed from the results section.

Figure 1C: THP-1 cells were transduced with LV shRNA-control or -RANBP2 or non-treated (NT) then infected with influenza A virus (A/WSN/1933) at MOI 0.5. At the indicated time points, cells were lysed and total intracellular RNAs were extracted. Viral RNAs encoding the Matrix 1 (M1) protein were amplified by RTqPCR.

Fig. 1G: NTD presumably stands for N terminal domain. This abbreviation was introduced in the introduction, but not in the figure legend. This should be corrected.

This has been amended. Now NTD refers to N-terminal domain and NT refers to non-transduced/non-treated.

Fig. 1I: Colocalization in the shRNA Control and RanBP2 groups look very similar. Some other way of visualization should be provided.

In both conditions, viral RNPs are exported, leading to the colocalization of cytoplasmic PB1 with NP. However, in the RANBP2-depleted cells, the cytoplasmic levels of NP are increased. This experiment aimed to determine whether the observed increase in NP corresponds to assembled RNPs or to unassembled, free NP protein.

Pearson's correlation coefficients were calculated across all experiments to quantify the degree of colocalization between NP and PB1 signals. As expected, these analyses confirmed colocalization. The figure panel has been updated to include colocalization curves, Pearson's coefficients determined across all three experiments, and zoomed-in views of the colocalization areas for improved clarity (see **Figure 1I-K, S5A-D**).

Fig. 1J: It is not detailed neither in the results section nor in the figure caption what is shown in the upper panel.

The upper panel showed an enlargement of the colocalization signal in the RANBP2-depleted cells. In the updated figure, the panel now includes zoomed-in regions of each condition, as well as Pearson's correlation coefficients. The figure legend has been revised as follows:

(lines 703-705): "(J) Colocalization curves and Pearson's correlation coefficients between PB1 and NP signals were quantified using the Fiji software. (K) Enlargements of colocalization fields from panel I."

In the caption of Figure 1, a typo should be correct (J).

The typo has been corrected.

In Fig. 3A it looks as if the analysis of XCXL10 transcripts in monocyte-derived macrophages has been performed only twice. How can statistics be performed with 2 values? This needs re-evaluation. The same is true for the analysis of TNFalpha in monocyte-derived macrophages. How exactly the experiment was carried out? When for the IL-1beta analysis in monocyte-derived macrophages three values were analyzed, then the samples should be basically available for the analysis of the other cytokine RNAs. In the experiment with THP-1 cells, treatment with control shRNA was also carried out only twice, whereas the experiment with RanPB2 was performed three times. This is no basis to perform statistics. On a first glance, only for the studies with A549 cells enough data points were retrieved to perform reasonable statistics. It is recommended that a professional statistician cross-checks the calculations.

Experiments were replicated and Figure 3A now includes A549 and U2OS cells. As the degree of activation varied between experiments, paired Student's t tests were performed (*P < 0.05, **P < 0.002, ***P < 0.0002, ****P < 0.0001, ns: non-significant).

For Fig. 3D more details should be provided. Was the experiment carried out only once?

Figure 3D does not exist. All assays were performed as three independent experiments, except the Luminex assay, which was conducted using primary cells from two donors due to cost constraints.

In Fig. 4A the caption should say: Schematic depiction of ... and in the figure this text should be deleted. Within this figure part, only one font size should be used. The labelling within the box is so small that it is hardly readable.

The figure legend for 4A was revised and font sizes in the panel have been increased and homogenized.

Fig. 4D: Sanger sequencing goes back to Fred Sanger, therefore it should be written with a capital first letter.

Sanger is now written with a capital letter on the figure.

In the legend of Fig. 4 reference is given to (I), but (I) is not shown in the figure. This should be corrected.

This is indeed an error that appeared during rewrites of the manuscript. The legend of figure 4 has been corrected.

In Fig. 4 too many different font sizes are used. In the end, every label is important, otherwise it would not have been shown. Therefore, the authors should consider using only one font size.

This has been corrected.

Reviewer #2 (Remarks to the Author):

Summary:

In this work, entitled 'The genetic driver of Acute Necrotizing Encephalopathy, RANBP2, regulates the inflammatory response to Influenza A virus infection', Desgraupes et al. describe that loss of RANBP2 leads to increased IAV vRNA replication and increased localization of vRNPs to the cytoplasm. This is accompanied with an increased pro-inflammatory cytokine response. CRISPR-edited cells expressing RANBP2 T585M, the SNP that is observed in ANE patients, also show increased CXCL10 expression in response to IAV infection or after stimulation with RIG-I 2CARD. Additionally, the T585M mutation disrupts the localization of RANBP2 at the nuclear pore. This study shows interesting insights into the function of RANBP2 in influenza virus infection, and how the RANBP2-T585M mutation could contribute to ANE. However, it is not clear if the observed increased inflammatory response in the absence of functional RANBP2 is caused by increased viral RNA replication or by exacerbated signaling downstream of RIG-I.

In the original manuscript, we included a single experiment using constitutive RIG-I, which suggested that the RANBP2 mutation increased CXCL10 (but not TNF α) expression downstream of RIG-I. Similar effects can be observed with other artificial innate immune triggers (LPS, R848, LV transduction). Indeed, in the absence of IAV, RANBP2 knockdown caused a modest yet significant increase in TNF α and IL-1 β levels (**Figure 3A-B**), suggesting that RANBP2 depletion may induce a mild pro-inflammatory state independently of viral replication (possibly due to LV transduction). This effect is low in magnitude and may only be observed in some cell types (e.g. U2OS). In contrast, IAV replication triggered a robust pro-inflammatory response in all tested cells (up to 12-14 fold increases in secreted CXCL8, CXCL10 and CCL3

proteins by primary macrophages), demonstrating RANBP2's role in controlling IAV-induced inflammation. Notably, unstimulated CRISPR clones did not show basal activation, unlike LV-transduced U2OS cells (**Figure 3A, Reviewer figure 1**).

Accordingly, we removed the RIG-I panel from the revised manuscript and replicated all key experiments using viral infection only. We also include a note in the text to acknowledge this observation:

(lines 196-199): "Interestingly, a modest increase in some of these transcripts was also observed in uninfected cells (**Figure 3A-B**), likely reflecting low level activation triggered by lentiviral vector transduction, and suggesting that RANBP2 knockdown may induce a mild pro-inflammatory state independently of its effect on IAV replication".

Reviewer Figure 1: Pro-inflammatory cytokine levels in unstimulated CRISPR clones.

CRISPR clones were lysed and total intracellular RNAs were extracted. Cytokine transcripts were amplified by RTqPCR.

Major comments:

1. All experiments shown in Figures 1-3 were done using only one shRNA. To exclude potential off-target effects of the used shRNA, key experiments need to be repeated with additional shRNAs and/or including rescue experiments (expression of shRNA resistant RANBP2).

In order to exclude potential off-target effects, we replicated all key experiments using an independent second shRNA targeting RANBP2 (position 225, **Figure S4A**) previously validated by Di Nunzio et al. (2012). Efficient RANBP2 knockdown was confirmed (**Figure S4B**). Consistent with the first shRNA, RANBP2 depletion caused cytoplasmic accumulation of NP, specifically in RANBP2-shRNA-expressing cells (white arrow heads, **Figure S4C** and **S4D**) and significantly increased CXCL10 (70-fold), TNFα (4-fold) and IL-6 (7-fold) following IAV infection, confirming its role in regulating the inflammatory response to IAV infection (**Figure S11**).

These results have been added to supplementary Figures S4 and S11, and the following sentences have been added in the results part:

(Lines 130-131): "This phenotype was confirmed using a previously published RANBP2-targeting shRNA (Di Nunzio et al., 2012; **Figure S4A-D**)."

(Lines 193-195): "RANBP2 depletion led to a significant upregulation of CXCL10, IL-6, TNF-α and IL-1β (**Figure 3A-B, Figure S10A-B**), which mirrored the increase in vRNA (**Figure S10C**) and was confirmed using an independent RANBP2-targeting shRNA (**Figure S11**)"

Figure S4: Increase in cytoplasmic NP in RANBP2-KD cells at 8 hpi, using a second shRNA

(A) Schematic depiction of the hybridizing sites of the two RANBP2-specific shRNAs. While the shRNA-RANBP2 #1 binds in the 3'UTR sequence, the shRNA-RANBP2 #2 binds at position 225 of RANBP2 RNA. (B) A549 cells were transduced with LV shRNA-control or shRANBP2 for 48h, and RANBP2 knock-down was verified by RTqPCR and Western Blot. (C) Transduced A549 cells were infected with influenza A virus (IAV) (A/WSN/1933) at MOI 0.5. At 8h post-infection, cells were fixed and stained for the nucleoprotein (NP). Scale bar: 20µm. (D) Cytoplasmic fluorescence was quantified using the Fiji software. Each dot represents the cytoplasmic fluorescence of a cell. Two-tailed unpaired Student's t test. ****P < 0.0001.

Figure S11: Increase in cytokine transcripts in RANBP2-KD cells, using a second shRNA

A549 cells were transduced with LV shRNA-control or shRANBP2 for 48h and infected with influenza A virus (IAV) (A/WSN/1933) at MOI 0.5. At 8h post-infection, cytokine transcripts were assessed by RT-qPCR. Results were normalized on RPL13a and on the stimulated control condition. Results are presented as mean +/- SD. Two-tailed unpaired Student's t test. *P < 0.05, **P < 0.002, ***P < 0.0002, ****P < 0.0001, ns: non-significant.

2. Figure S1: The western blot shows the knockdown at protein level in A549 cells at 72h post lentiviral transduction. Is this the time point that was used for infection with IAV? I could not find this information in the method section. The authors should add this information and potentially provide a western blot for the time point of infection.

For experiments in A549, infection was initiated at 48 h post-transduction but all experiments were ended between 56-96 h post-transduction, which is why we showed knockdown at 72 h.

The revised figure now includes western blots at both 48 h and 72 h post-transduction with quantification of 3 independent blots (**Figure S1A-B**). We also included this information in the method section:

(lines 348): “All experiments were completed by 56-96 h post-transduction.”

Figure S1: Assessment of RANBP2 knock-down in A549 cells

A549 cells were transduced with LV shRNA-control or shRNA-RANBP2 at MOI 20. **(A)** Knock-down efficacy was assessed by western Blot at 48 h and 72 h post-transduction in A549 cells. **(B)** Quantification of band intensity was done using the ImageLab software. Results are normalized on the vinculin signal and on the control condition.

The western blot showing RANBP2 reduction at protein level should also be added for MDM cells (where currently only a qPCR at 24h is shown).

RANBP2 protein levels were assessed in MDMs at 72 h post-transduction. A representative blot is now included, together with a quantification from 3 independent experiments (**Figure S14B**).

Figure S14B: Transduction with LV shRNA-RANBP2 leads to an efficient down-regulation of RANBP2 in MDM cells. MDM cells were differentiated from primary monocytes and transduced with LV shRNA-Control or shRANBP2 for 72 h. RANBP2 knock-down was verified by Western Blot. Quantification of band intensity was done using the ImageLab software. Results are normalized on the vinculin signal and on the control condition.

3. Fig 1B and Fig 3A: The data showing the knockdown efficiency of shRNA-transduced THP1 cells is missing. The authors should provide a western blot showing RANBP2 levels like for the A549 cells in Figure S1B.

RANBP2 protein levels were assessed in THP-1 cells at 72 h post-transduction (**Figure S1C**).

Figure S1C: Transduction with LV shRNA-RANBP2 leads to an efficient down-regulation of RANBP2 in THP-1 cells. THP-1 cells were transduced with LV shRNA-Control or shRANBP2 for 72 h. RANBP2 knock-down was verified by Western Blot.

4. The authors state that nuclear export of segment 1, 2, 3, 7 and 8 is increased by RANBP2 KD. I find it difficult to see this in the data shown in Figure 2F. For all segments, the cytoplasmic levels seem to be increased to at least some extent, and the nuclear levels are either equal or slightly increased compared to the control. The increase of RNA levels in the cytoplasm in RANBP2 KD compared to control is between 1.5-3-fold for all segments, which is a difference of 2-fold in-between the different segments. In a qPCR assay, this change corresponds to only 1 Cycle difference. I think that from this data it is difficult to conclude that the different segments are actually affected differently by RANBP2 KD. Additionally, I wonder why there is no decrease in the nuclear RNA levels observed if nuclear export is increased.

We agree that the evidence for differential vRNP segment nuclear export needed to be strengthened and required clearer presentation. During the revision, we repeated the experiment multiple times and consistently observed increased vRNA export upon RANBP2 knockdown, though the specific segments affected varied between replicates. The revised data (Figures 2H-J) now show that RANBP2-KD causes over-export of most vRNAs, unevenly affecting individual segments and thereby altering segment stoichiometry. Consistent with this, nuclear vRNA levels decrease when export is increased (Figure 2H).

The manuscript text was also revised as follows:

(lines 178-183): “RANBP2 knockdown ... led to an overall increase in the nuclear export of vRNA segments (Figure 2H-I). This effect disproportionately affected individual segments across experiments, as reflected by a higher standard deviation (Figure 2J). These findings indicate that, while the depletion of RANBP2 increases the replication of all vRNA segments, it also promotes their uneven export to the cytoplasm, thereby disrupting segment stoichiometry.”

Figure 2: (H) Transduced A549 cells were infected with IAV at MOI 0.5. At 6h post-infection, nuclear and cytoplasmic fractions were separated and total intracellular RNAs were extracted. vRNAs were reverse transcribed by strand-specific RTqPCR and for each segment, the cytoplasmic and nuclear abundances were expressed as percentages of their sum. Technical triplicates from a representative experiment are presented. (I) Percentage of vRNAs exported to the cytoplasm (all segments pooled), from two independent experiments. Each dot represents the mean of three technical replicates per segment. Paired analysis was performed using the

Kolmogorov-Smirnov test. **P < 0.002. (J) Same data as (I), shown as mean +/- range. Standard deviation values were determined for each condition. Kolmogorov-Smirnov test. ****P < 0.0001.

5. While all virological assays were done in A549 cells, the experiments investigating the effect of RANBP2-T585M (Figure 4), were done in U2OS cells. It is not clear why the authors switch to a different cell line. I assume this is for technical reasons, as it might be difficult to do CRISPR editing in A549 cells. I think the authors should reproduce some of the shRNA KD data (increased cytoplasmic export of vRNPs and increased cytokine production in RANBP KD conditions) in U2OS cells to show if this cell model shows the same phenotype as A549 cells. Additionally, the increase in cytokine induction appears much stronger in U2OS RANBP2-T585M cells (Figure 4E) compared to A549 RANBP2 KD cells (Figure 3A). Is this due to differences in the cell lines, or does the T585M mutation increase cytokine production compared to RANBP2 KD?

To reproduce the virological assays in the same cell line used for CRISPR editing, we depleted RANBP2 in U2OS cells (Figure S10A-B) and assessed both the viral phenotype (Figure S10C) and cytokine production (Figure 3B). RANBP2 knockdown increased viral RNA levels, and elevated CXCL10, TNF α , IL-6 and IL-1 β transcripts, confirming the phenotype observed in A549 cells. Moreover, although stimulation of both cell types induced similar cytokine levels in control cells, higher induction was observed in RANBP2-depleted U2OS cells, suggesting that the higher increase observed in CRISPR clones likely reflects U2OS-specific responses rather than the T585M mutation itself.

Figure S10: RANBP2 knockdown in U2OS cells leads to increased IAV RNA

Transduced A549 cells were infected with IAV at MOI 0.5. RANBP2 knockdown was quantified by (A) RTqPCR and (B) western blot, at 72 h post-transduction. (C) Total intracellular RNAs were extracted and viral RNAs encoding the Matrix 1 (M1) protein were amplified by RTqPCR. Results are normalized on RPL13a and on the control condition (mean +/- SD). Each dot represents a technical replicate from three independent experiments. Two-tailed unpaired Student's t test, ****P < 0.0001.

Figure 3B: U2OS cells were transduced with LV shRNA-control or shRNA-RANBP2 at MOI 30 and stimulated with influenza A virus (IAV) (A/WSN/1933) at MOI 0.5 overnight. Cytokine transcripts were assessed by RT-qPCR and results were normalized on RPL13a and on the unstimulated control condition. Results are presented as mean +/- SD. Three independent experiments. Two-tailed paired Student's t tests. *P < 0.05, **P < 0.002, ***P < 0.0002, ****P < 0.0001, ns: non-significant.

6. In Figure 4, the authors show an effect on CXCL10 with RIG-I 2CARD expression in RANBP2-T585M cells. Does this mean that the phenotype observed in cells expressing RANBP2-T585M is a different one than observed in the KD cells, caused downstream of RIG-I sensing and not by increased viral RNA levels or changed vRNA localization? To clarify this, I would suggest to test if import of viral polymerase and vRNP export are affected in U2OS RANBP2-T585M cells. And vice versa, could the authors repeat the experiment shown in Figure 3A with RIG-I 2CARD transfection instead of IAV infection?

As mentioned above (see comment to Summary above), the use of synthetic agonists such as constitutive RIG-I can lead to low level background activation. Indeed, in the absence of IAV, RANBP2 knockdown causes a modest yet significant increase in TNF α and IL-1 β levels (revised **Figure 3B**), suggesting that RANBP2 depletion may induce a mild pro-inflammatory state independently of viral replication (either through an additional mechanism or possibly as a result of lentiviral vector transduction). Nevertheless, despite this mild basal activation, IAV replication triggers a considerable outburst in pro-inflammatory cytokines, clearly highlighting the role of RANBP2 in controlling IAV-induced inflammation.

In contrast, RANBP2-T585M did not induce higher basal levels of pro-inflammatory cytokines in unstimulated clones (see **Reviewer Figure 1** below), suggesting a specific mechanism triggered by IAV in ANE1 mutant cells.

Reviewer Figure 1: Pro-inflammatory cytokine levels in unstimulated CRISPR clones.

CRISPR clones were lysed and total intracellular RNAs were extracted. Cytokine transcripts were amplified by RTqPCR.

Regarding viral RNP export, we have assessed whether this process is impacted in mutant CRISPR clones. Remarkably, while vRNPs remained nuclear in WT clones, cells harboring the T585M mutation of RANBP2 displayed increased cytoplasmic NP localization (**Figure 4E-F**), accompanied by increased vRNA replication (**Figure 4G**).

These data have been included in Figure 4 of the revised manuscript and the text has been adapted accordingly in the result section.

(Lines 232- 236): “CRISPR-knock-in clones were infected with IAV, and replication was assessed by quantifying viral NP at 8 hpi, as previously (Figure 1G-H). Strikingly, cytoplasmic NP was significantly increased in all ANE1 CRISPR clones compared with wild-type clones, regardless of whether the mutation was heterozygous or homozygous (Figure 4E-F, Figure S16). This was accompanied by elevated vRNA levels, which was statistically significant at 24 hpi (Figure 4G).”

Figure 4E-G: The RANBP2 T585M mutation leads to increased NP export and IAV RNA levels.

CRISPR clones were stimulated by IAV (A/WSN/1933) at MOI 0.5. **(A)** At 8h post-infection, cells were fixed and stained for the nucleoprotein (NP). Scale bar: 20µm. **(B)** Cytoplasmic fluorescence was quantified using the Fiji software. Multiple Welch’s t tests. *P < 0.05, **P < 0.002, ***P < 0.0002, ****P < 0.0001, ns: non-significant. **(C)** At 1h, 8h and 24h post-infection, total intracellular RNAs were extracted and viral RNAs encoding the Matrix 1 (M1) protein were quantified by RTqPCR. For each clone, results were normalized on RPL13a and on the early time point (1h). One-way ANOVA. *P < 0.05, **P < 0.002, ***P < 0.0002, ****P < 0.0001, ns: non-significant.

7. Are the protein levels of RANBP2-T585M the same as WT RANBP2 in U2OS cells? It would be important to show this, to exclude that the observed phenotype of the T585M mutation is not caused by reduced protein levels.

We determined RANBP2 protein levels by western blot in CRISPR clones and did not observe any statistically significant variations.

This data has been added to supplementary Figure S17 and the following sentence has been added in the text: (Lines 245-246): “Overall, we did not observe a statistically significant variation in the protein levels of RANBP2 (Figure S17A-B).”

Figure S17: RANBP2 protein levels in CRISPR clones

(A) CRISPR clones were lysed and protein levels of RANBP2 were determined by western blot. (B) Quantification of band intensity was done using the ImageLab software. Results are normalized on the vinculin signal and on the C4 WT clone. One-way ANOVA. ns: non-significant.

Minor comments:

- What is the unit of the MOIs stated in the manuscript? PFU/cell or TCID50/cell?

The MOI is indicated as TU/cell. The information has been added in material and methods line 345.

- I think the qPCR primers used for the validation of strand-specific RT shown in Fig S2 are not listed in the supplement. If I understood correctly, these primers are different from the qPCR primers used in the other assays, since they are specific for the tags added during the RT?

These primers were indeed overlooked. They have now been added to the primer table (Table S2) of the material and method section. Additionally, we noticed that the 5'/3' endings were inverted on the scheme of supplementary figure S2 (for cRNA, lower box), this has been corrected on the figure.

- Figure 1J: From which condition is the magnified example image and the analysis? Could the authors provide this analysis showing the signal distribution for all conditions? Could the authors additionally provide a quantification of the colocalization (e.g. pearson coefficient) for all conditions?

The upper panel (**Former Figure 1J**) showed an enlargement of the colocalization signal in the RANBP2-depleted cells. In the updated figure, the panel now includes zoomed-in regions of each condition, as well as Pearson's correlation coefficients (**Updated Figure 1I-J-K and Figure S5A-D**).

In both conditions, viral RNPs are exported, leading to the colocalization of cytoplasmic PB1 with NP. However, in the RANBP2-depleted cells, the cytoplasmic levels of NP are increased (**Figure 1G-H**). This experiment aimed to determine whether the observed increase in NP corresponds to assembled RNPs or to unassembled, free NP protein.

Pearson's correlation coefficients were calculated across all experiments to quantify the degree of colocalization between NP and PB1 signals. As expected, these analyses confirmed colocalization. The figure panel has been updated to include colocalization curves, Pearson's coefficients, and zoomed-in views of the colocalization areas for improved clarity.

Figure 1I-J-K: RANBP2 depletion leads to an increase in viral RNP export.

A549 cells were transduced with LV shRNA-Control or shRANBP2 and infected with influenza A virus (IAV) (A/WSN/1933) at MOI 0.5. **(I)** Transduced A549 cells were infected at MOI 0.5 and co-stained for the NP and PB1 proteins at 8h post-infection. Scale bar: 20 μm . **(J)** Colocalization curves and Pearson's correlation coefficients between PB1 and NP signals were quantified using the Fiji software. Welch's t test. * $P < 0.05$, ** $P < 0.002$, *** $P < 0.0002$, **** $P < 0.0001$, ns: non-significant. **(K)** Enlargements of colocalization fields from panel I.

Figure S5: Increase in cytoplasmic vRNP in RANBP2-KD cells at 8 hpi

Transduced A549 cells were infected at MOI 0.5 and co-stained for the NP and PB1 **(A)** or for the NP and PA **(C)** proteins at 8h post-infection. Scale bar: 20 μm . **(B-D)** Pearson's correlation coefficients between both signals were quantified using the Fiji software.

- Number of biological replicates is not stated for the experiments. Especially for the western blots shown in Fig 2A and 2B, how often was this experiment done?

Western blots in Figure 2A-B were performed and quantified in three independent experiments. Western blots in Figure 2C-D were performed five times. Nuclear fractions were pure in all experiments (quantifications were made across all five experiment), but cytoplasmic fractions were successfully purified twice.

The information about the number of replicates has been added to all figure legends.

- Figure 2E: It is not clear to me why the authors analyze nuclear export of cRNAs. As far as I know, cRNAs would not need to be exported to the cytoplasm during the IAV life cycle, since they serve as template for vRNA synthesis that occurs in the nucleus.

To meet this reviewer's comment, the cRNA data has been moved to supplementary figures (**revised Figure S9**). However, we have retained this data in the manuscript, as it provides relevant biological information regarding the nuclear increase of all cRNA segments and constitutes an important control of the fractionation efficacy.

- Figure 2E and F: How many replicates were done of this experiment? Does each dot represent a technical replicate from the same experiment or a biological replicate from a different experiment? Please add this information to the figure legend. Additionally, the statistical analysis for this experiment should be added.

In the revised manuscript, these panels have been changed. Figure 2E is now Figures 2H-J and Figure 2F is now Figure S9.

To address this reviewer's concern, we repeated the experiment multiple times and consistently observed increased vRNA export upon RANBP2 knockdown, though the specific segments affected varied between replicates. Since vRNP segment nuclear export required clearer presentation, panel 2H shows technical triplicates from a representative experiment, while panels 2I, 2J and S9 are the mean of technical triplicates from two or three independent experiments.

We have added the information about the number of replicates and statistical analysis to all figure legends.

- In the qPCRs in Figure 2E and F, the data was normalized in different ways. In 2E (cRNA), normalization is relative to the nuclear fraction of the control, which means that nuclear and cytoplasmic fractions can be compared to each other, and it is apparent that most of the cRNA is located to the nucleus. In Figure 2F, the same experiment is shown for vRNA, but here the fractions are normalized to the control condition within each fraction. Thus, it is not apparent how much RNA is exported to the nucleus at the observed time point. It would be better to do the normalization in the same way as for Figure 2E.

This panel was not retained in the revised manuscript (see comment above).

- Line 253: 'In particular, RANBP2 was not involved in the initial export of viral transcripts, which involves the general mRNA export pathway via NXF1:NXT' – Did the authors test this? Please show data on viral mRNAs, or remove this statement from the discussion.

The export of influenza virus messenger RNAs has been extensively documented (Refs 1-4 below), and is known to rely on the NXF1:NXT1 pathway. Consistent with this, in our

experiments, inhibiting CRM1 did not impact the translation of PB1, PB2 and PA (data not shown), suggesting that viral mRNA export is not affected by RANBP2-KD. **We have reformulated the text as follows:**

(Line 273): “and did not seem to affect the initial export of viral transcripts, which involves the general mRNA export pathway via NXF1:NXT1.”

(1) Bhat, P., Aksenova, V., Gazzara, M. et al. **Influenza virus mRNAs encode determinants for nuclear export via the cellular TREX-2 complex.** Nat Commun 14, 2304 (2023). <https://doi.org/10.1038/s41467-023-37911-0>

(2) Zhao L, Liu Q, Huang J, Lu Y, Zhao Y, Ping J (2022) **TREX (transcription/export)-NP complex exerts a dual effect on regulating polymerase activity and replication of influenza A virus.** PLoS Pathog 18(9): e1010835. <https://doi.org/10.1371/journal.ppat.1010835>

(3) Ke Zhang, Tolga Cagatay, Dongqi Xie, Alexia E. Angelos, Serena Cornelius, Vasilisa Aksenova, Sadaf Aslam, Zhiyu He, Matthew Esparza, Ashley Vazhavilla, Mary Dasso, Adolfo García-Sastre, Yi Ren, Beatriz M.A. Fontoura, **Cellular NS1-BP protein interacts with the mRNA export receptor NXF1 to mediate nuclear export of influenza virus M mRNAs,** Journal of Biological Chemistry, Volume 300, Issue 11, 2024, 107871, ISSN 0021-9258, <https://doi.org/10.1016/j.jbc.2024.107871>

(4) Pereira CFRead EKCWise HMAmorim MJ, Digard P2017. **Influenza A Virus NS1 Protein Promotes Efficient Nuclear Export of Unspliced Viral M1 mRNA.** J Virol91:10.1128/jvi.00528-17. <https://doi.org/10.1128/jvi.00528-17>

Comments on Methods part.

The following information missing in the methods part:

- The amounts of plasmids transfected to produce lentiviral particles.

The information has been added in the material & methods section:

(lines 333-334): “Lentiviral vectors (LV) were produced in HEK-293T cells (ATCC) by calcium phosphate transfection using a VSV-G envelope plasmid (pVSVg, **15 µg** in a 70-80% confluent 175cm² flask), a Gag-Pol encapsidation plasmid (p8.74, **30 µg**), and the transfer plasmid of interest (**30 µg**; mCherry-shRNA-control, mCherry-shRNA-RANBP2 or Centauri-NLuc-NLS; see Table S1).”

- Detailed description of how A549 cells were transduced with the lentiviral particles. It is not clear from the manuscript if there was a selection step and if stable cell lines were produced, or if the A549 cells were freshly transduced and immediately infected with IAV in each experiment.

A549 cells were freshly transduced and we do not use selection since transduction efficiencies are consistently >95% in cell lines. This information has been added to the material & methods section.

(Lines 343- 348): “Cells were seeded into plates one day prior to transduction. After a PBS wash, lentiviral transduction was performed at MOI 20 to 40 (transducing units/cell) in a small volume of culture medium supplemented with 2% FBS, to facilitate viral adsorption. After 2 h of incubation at 37°C, 10% FBS culture medium was added. Transduced cells were cultured for 48-72 h prior to viral infections to allow efficient knock-down. All experiments were completed by 56-96 h post-transduction”.

- Information on IAV-WSN stock production and stock titration.

This information has been added to the material & methods section.

(Lines 349- 353): “H1N1/A/WSN/1933 viral stocks were produced by amplification at an MOI of 10⁻⁴ on MDCK cells, cultured in Minimal Essential Medium (MEM, Gibco) containing 2%-FBS and 1 µg/mL of TPCK-treated trypsin for 3 days at 37°C. Viral supernatants were harvested and centrifuged for 5 min at 2,000 g to remove cellular debris. Aliquots were stored at -80°C. IAV titers were determined by TCID₅₀ assay as described below.”

- Line 315: ‘For experiments with NF-κB canonical pathway inhibition, cells were pre-treated with 2% FBS culture medium containing 20 µM of PS1145 (MedChemExpress) for 1 h at 37°C’. It is not clear which experiments were performed with inhibition of NF-κB. These experiments were not included in the original manuscript and this sentence has been removed.

Reviewer #3 (Remarks to the Author):

The submitted study by Desgraupes et al. aims to understand the role of RANBP2 during influenza A virus infection. Through their interesting work, these authors were able to show that RANBP2 is able to directly influence the inflammatory response of epithelial and immune cells during infection. The manuscript is extremely well-written and their experimental work is very close-knit. This study provides significant groundwork towards the understanding of the exact mechanisms of RANBP2, especially in the context of patients with the genetic predisposition to developing ANE.

Major comments:

- Lines 183-185, there is an inherently large jump here from the previous section. Some improvement for the motivation here to align the two sections is missing.

The transition between the two sections was re-written as follows.

(Line 192-194): “To investigate whether the abnormal cytoplasmic accumulation of vRNA, observed in RANBP2-depleted cells, might associate with increased inflammation, we measured the expression of inflammatory transcripts by qPCR.”

And in the following paragraph (Lines 200-201): “To confirm this at the protein level, primary monocytes were isolated from PBMCs from healthy donors and differentiated into pro-inflammatory M1-like macrophages.”

Moreover, it was not established if the difference in vRNAs and vRNPs dynamics were also observed in these MDMs, before looking into the expression of proinflammatory genes.

While we agree that demonstrating the vRNA and vRNP dynamics in MDMs would strengthen the manuscript, our attempts to do so using the same conditions applied to A549, THP-1 and U2OS cells were not successful. Primary MDMs exhibited high background in imaging assays, and infection efficiency at the single-cell level was more variable across donors than in immortalized lines, preventing reproducible quantification of vRNA abundance and vRNP localization. In addition, RANBP2 knockdown in MDMs, although satisfactory at the transcriptional level (~60% inhibition, **Figure S14A**), was modest at the protein level (~40% inhibition, **Figure S14B**). This is likely due to the non-cycling nature of MDMs, which limits the turnover of existing protein pools and leads to increased RANBP2 half-life and nuclear pore residency. Despite these technical constraints, MDMs displayed a robust increase in pro-inflammatory gene expression after IAV exposure (**Figure 3C**), consistent with the phenotype

observed in all the other cell types examined. Similarly to the WT/T585M heterozygous clone, where subtle alterations in RANBP2 localization at the nuclear rim (**Figure 4K**) produce major changes in the inflammatory response (**Figure 4H**), these findings indicate that even minor perturbations in RANBP2 can lead to substantial IAV-induced inflammatory effects, underlining the relevance of these results to the ANE1 physiopathology.

- The selection of the respiratory epithelial cell line A549 cells (adenocarcinoma) for the initial investigation makes sense. However, this reviewer would like to see the dynamics of proinflammatory gene expression after infection in upper respiratory cells to explicitly show the effect of the RANBP2 in IAV appropriate target cells. This would make an interesting comparison with the effect of infection of MDMs in this study (epithelial vs. immune proinflammatory responses).

We appreciate the reviewer's suggestion to include experiments using primary upper respiratory tract models. *In vitro* studies of the respiratory epithelium can indeed be performed using air-liquid interface (ALI) cultures or 3D organoids. However, due to budgetary and time constraints, we were unfortunately unable to invest in these approaches. More importantly, while analyzing proinflammatory gene expression in upper respiratory cells could provide additional insight into local host responses to IAV infection, this approach is not directly relevant to the ANE1 pathology. ANE1 is a neurological disorder that primarily affects the brain, leading to acute encephalopathy and hallmark bilateral lesions of the thalami. Current evidence suggests that the neurological damage is not driven by local effects at the primary site of infection but by downstream effects on the central nervous system.

- Figure 3, the authors show that there are differences in the gene expression pattern of these proinflammatory markers after IAV infection after an overnight infection strategy. This reviewer thinks that a variable timelapse analysis may be necessary to clearly say that the differences in gene expression are a result of the cell types. It could be possible that some gene expression changes had happened a lot earlier in some cells and later in others. A time-resolved infection strategy with a qRT-PCR readout would help here.

As noted by Reviewer 1, this panel lacked sufficient experimental replicates for some cell types (see comment above "In Fig. 3A it looks as if the analysis of XCXL10 transcripts in monocyte-derived macrophages has been performed only twice."). We have therefore removed this panel from the revised manuscript. Instead, **Figure 3A-B** now include triplicate experiments from A549 and U2OS cells, which show similar results, thus arguing against a cell-type-dependent effect.

- Have the authors looked into interferon stimulated genes (ISGs) and the influence of RANBP2 here?

Although we have not specifically analyzed interferon-stimulated genes (ISGs), our data do not support a strong involvement of RANBP2 in the interferon (IFN) pathway. First, RANBP2 expression itself is not responsive to IFN α stimulation (see **Reviewer Figure 2**), suggesting it is not an ISG. Second, we did not observe major alterations in key components or outputs of IFN signaling following RANBP2 knockdown. For example, IFN α induction in MDMs was modest (**Figure 3C**), and IFN β levels are not increased in RANBP2-depleted cells upon IAV infection (**Reviewer Figure 3**). Together, these observations indicate that, under our experimental conditions, RANBP2 depletion does not markedly influence IFN production or signaling responses.

Reviewer Figure 2: RANBP2 expression is not modified by IFN α stimulation.

Jurkat, CEM or THP-1 cells (treated or not with 100 ng/mL PMA) were stimulated with 100 U/mL IFN α for 16 h. Total intracellular RNAs were extracted and RANBP2 expression was quantified by RTqPCR.

Reviewer Figure 3: RANBP2-depletion does not modify IFN β expression.

A549 cells were transduced with LV shRNA-control or shRANBP2 for 48h and infected with influenza A virus (IAV) (A/WSN/1933) at MOI 0.5. At 16h post-infection, cytokine transcripts were assessed by RT-qPCR. Results were normalized on RPL13a and on the stimulated control condition. Results are presented as mean +/- SD. Two-tailed unpaired Student's t test. *P < 0.05, **P < 0.002, ***P < 0.0002, ****P < 0.0001, ns: non-significant.

Minor comments:

- In line 105, the figures do not match the cell lines in the text. The A549 and THP-1 cells were inadvertently inverted. This has now been corrected as follows (line 111): "In A549 cells (Figure 1B) and in the monocytic cell line THP-1 (Figure 1C, Figure S1)."
- Lines 178-180, well-written but must be revised to be more explicit that this is still heavily speculative at this point. The title for this results section has been revised to be more explicit as follows: (line 171-172): "RANBP2 depletion favours the export of vRNA segments to the cytoplasm and disrupts the segment stoichiometry".
- Figure 4D, please add the clone number on the traces to make it easier to read. The modification has been made on the figure.
- Line 765, please use the proper β symbol for IL-1 β . This has been amended (line 775).

REVIEWER COMMENTS

Reviewer #1 (Remarks to the Author):

Desgraupes et al. followed many of the reviewers' recommendations and implemented several changes in the revised version of their manuscript. New controls were included, and experiments were repeated where needed in order to be able to provide statistics. Thanks to the extended discussion, more context is provided. However, in all figures still different font sizes are used which makes it difficult to read the figures. Since many readers certainly will want to verify some of the conclusions by inspection of the actual data, the authors should try to present the data in an easily readable manner.

We thank the reviewer for their constructive feedback and have harmonised the font sizes across figures (main and supplemental) to improve readability. All labels - including axes, schematics, stainings, statistics, etc – are now in font 9.

Minor points:

Abbreviations in context of RNA are inconsistently used. In line 47 of the revised manuscript, it is defined that vRNA stands for "negative-sense single-stranded viral RNA". However, many times later in the manuscript the authors again refer to „viral RNA“. Do they use "vRNA" and "viral RNA" in a synonymously manner? If this is the case, the authors should better stay with the abbreviation vRNA throughout the manuscript after introduction the abbreviation.

In this manuscript, the abbreviation vRNA (viral RNA) refers exclusively to the negative-sense genomic RNA, while "viral RNAs" (plural, non-abbreviated) denotes all RNA species derived from the virus, including vRNA, cRNA and mRNA. We have checked the manuscript and confirm that the correct terms and abbreviations are used throughout.

shRNA is not defined in the manuscript. Considering that it is mentioned more than 100 times in the text, the authors should introduce the abbreviation, which presumably stands for "short" or "small hairpin RNA".

This was introduced line 102.

The expression „pro-inflammatory M1-like macrophages“ can be misleading. It is recommended that the authors better refer to "GM-CSF macrophages".

This change was made line 201.

Reviewer #2 (Remarks to the Author):

In the revised version of the manuscript 'The genetic driver of Acute Necrotizing Encephalopathy, RANBP2, regulates the inflammatory response to Influenza A virus infection' the authors improved the initial manuscript. Importantly, they added RANBP2 knockdown data in U2OS cells, the cell line which is also used to characterize the ANE-associated T585M mutation, and confirmed the observed phenotype using a second shRNA for RANBP2 knockdown.

The authors removed the RIG-I data, which was rather weak in the previous version. In the revised manuscript, the authors do not show a link between increased cytoplasmic vRNA and RIG-I sensing, but I feel it is justified to make this connection from the literature.

In the revised version of the manuscript, the authors addressed all of my concerns appropriately. I have two comments regarding the presentation of the data:

- It is difficult to compare the data in Fig 3A and Fig S11, which show cytokine expression for RANBP2 knockdown with two different shRNAs in A549 cells, since the normalization of the data was performed differently. Especially for CXCL10, it looks like with shRNA#1 (Fig 3A), the induction is ~2

fold (IAV infected, control vs. RANBP2 shRNA), while with shRNA#2 (Fig S11), it is more than 50-fold. Is there a way to improve the visualization to make it easier to compare the two different shRNAs? We thank the reviewer for this comment. In A549 cells, RANBP2 knockdown consistently leads to a 2- to 5-fold increase in TNF α , IL-6, and IL-1 β following IAV infection, whereas CXCL10 changes range from 2-fold to 50-fold. This may be explained by variable basal levels of CXCL10 between cell batches, cell passages, or slight variability in IAV inoculum, as CXCL10 mRNA is highly responsive to viral replication and interferon signaling. Thus, while we can confidently conclude that the effect on inflammation is RANBP2-specific, the difference in CXCL10 induction between the two shRNAs likely reflect experimental variability rather than mechanistic differences. For this reason and because, in the previous revision, reviewers asked to see unstimulated cells, we would prefer to retain the current representation of Figure 3A.

- Fig 2D: Figure legend states n = 5, but for PA only 4 data points are shown. Can the 5th experiment also be included?

We thank the reviewer for noticing this. The 5th data point has now been added to the figure (Fig 2D).

Reviewer #3 (Remarks to the Author):

The revised study by Desgraupes et al. is a significant improvement from the prior submission. This reviewer thanks the authors for the additional data they have provided (in the manuscript and those in the rebuttal only).

Major comment:

There is only one point that this reviewer would like to raise which is the confidence on the inflammatory effect of RANBP2. The hint of its presence is there but the wide spread of measurements in Fig3A and Fig4G can raise some concerns.

Fig 3A qPCRs - measurements for control and RANBP2 have a high spread and most significant changes are being driven by 1 measurement point. Have these been tested for outliers or maybe the experiment needs to be repeated?

Fig4G- needs the dots on the box whisker plots. The spread of these measurements are extremely high. The hint of the effect is there but additional confirmation of this would be nice. Similar to Fig3A, test for outliers and a potential repeat of the experiment might be necessary.

We appreciate the reviewer's observation regarding the variability in qPCR measurements. While there is indeed a wide spread, we believe the increased inflammatory effect observed upon IAV infection in RANBP2 knockdown cells is robust. In all experiments performed with RANBP2 knockdown, we consistently observed an increase in pro-inflammatory cytokine synthesis - across different cell types (A549, U2OS, THP-1, and primary macrophages), different experimenters, different lentiviral stocks, and supported by statistical analysis. The reviewer's impression likely arises from the fact that the magnitude of cytokine induction varies between experiments in the control conditions (see reviewer Figure 1 below), which makes the pooled data appear less striking. This biological variability can be explained by:

(i) Differences in basal cytokine levels between cell batches or passages. Since data are normalized to the control condition, low starting expression can make fold changes appear larger (see comment to Reviewer 2).

(ii) Differences in cell density. Previous work has demonstrated that cell activation and cytokine expression vary considerably with cell density, not only at the time of plating but also based on the cumulative history of density during passaging, thus creating heterogeneity in cytokine expression between experiments (Muldoon et al., Nat Commun 2020, PMID: 32054845; Binan et al., Cell 2025, PMID: 40081369). Generally, cytokine expression increases with confluency, a trend we have also observed for IL-1 β and IL-6 (data not shown). Therefore, variations in cell density at the time of RNA

extraction, as well as throughout passing and preparation, are likely to account for the observed variability.

Therefore, we prefer not to remove any outliers, as they do not represent technical errors but rather biological variability, and still allow the reader to see the effect of RANBP2-KD.

Reviewer figure 1: Reproducible increase of cytokine expression in the RANBP2-KD condition across experimental replicates. A549 cells were transduced with LV shRNA-control or shRANBP2 for 48h and infected with influenza A virus (IAV) (A/WSN/1933) at MOI 0.5 overnight. Cytokine transcripts were assessed by RT-qPCR. Results were normalized on RPL13a and on the unstimulated control condition (mean +/- SD). Single experiments from Figure 3A are presented (N1-N4), with each dot representing a technical replicate.

Fig3C - LUMINEX assays do not show TNFα, IL6 and IL1b increase and only for CXCL10 which is different from what was found in A549 cells and U2OS. Any point of discussion here?

We appreciate the reviewer's observation regarding the Luminex results.

Several factors may contribute to these differences, including the fact that secreted protein levels do not necessarily mirror transcriptional output, and that different cell types produce distinct cytokine profiles. Fig 3C presents fold changes relative to the infected shControl condition. Accordingly, our data in non-immune and immune cell types show increases in different cytokines, yet consistently confirm the reproducible effect of RANBP2 on inflammation.

In response to this reviewer's point, **we have added a sentence to the discussion to acknowledge this** (line 289-293): "Although there was considerable variability in the magnitude of this effect across experiments, as well as differences in the cytokines induced, particularly when comparing non-immune cell lines (A549, U2OS) with primary immune cells (macrophages), the global increase in inflammatory activation upon RANBP2-KD was consistently observed."